# Evaluation of a digital entomological surveillance planning tool for malaria vector control: Three country mixed methods pilot study

**Charlotte Hemingway**[1]*, **Steven Gowelo**[2], Mercy Opiyo[2], Dulcisaria Marrenjo[3], Mara Maquina[4], Blessings N. Kaunda-Khangamwa[5,6], Lusungu Kayira[5], Teklu Cherkose[7], Yohannes Hailemichael[7], Neusa Torres[4], Estevao Mucavele[4], Muanacha Mintade[4], Baltazar Candrinho[3], Themba Mzilahowa[5], Endalamaw Gadisa[7], Allison Tatarsky[2], Élodie A. Vajda[2], Emily Dantzer[2], Edward Thomsen[2], Michael Coleman[1‡], Neil F. Lobo[1,8‡]

1 Department of International Public Health, Liverpool School of Tropical Medicine, Liverpool, United Kingdom, 2 Malaria Elimination Initiative, University of California, San Francisco, San Francisco, California, United States of America, 3 Programa Nacional de Controlo da Malária, Maputo, Mozambique, 4 Centro de Investigação em Saúde de Manhiça, Mozambique, 5 Malaria Alert Centre, Communicable Diseases Action Centre, Kamuzu University of Health Sciences, Blantyre, Malawi, 6 School of Public Health, The University of the Witwatersrand, Johannesburg, South Africa, 7 Armauer Hansen Research Institute, Addis Ababa, Ethiopia, 8 Department of Biological Sciences, University of Notre Dame, Notre Dame, Indianna, United States of America

☯ These authors contributed equally to this work.
‡ MC and NFL also contributed equally to this work.
* charlotte.hemingway@lstmed.ac.uk

## Abstract

### Background

Vector control remains the principal method to prevent malaria transmission and has led to significant reductions in malaria incidence across endemic regions. However, such gains have stagnated, underscoring the need to tailor vector control to local drivers of transmission. An Entomological Surveillance Planning Tool (ESPT) was designed to translate normative guidance into an operational tool that supports cost effective, locally tailored, and evidence-based vector control. To facilitate ESPT implementation, an interactive digital toolkit (eSPT) was created to support question-based surveillance planning.

### Methods

The eSPT was evaluated with 49 target users in Ethiopia, Malawi, and Mozambique. The eSPT was introduced to participants through facilitated workshops. A mixed-methods design was employed, combining pre- and post-intervention surveys with qualitative measures to assess the impact of the eSPT on knowledge, self-efficacy and work practices related to entomological surveillance planning. Qualitative methods were used to explore the acceptability and utility of the eSPT.

**Data availability statement:** Anonymized response data from the knowledge and self-efficacy questionnaire is provided in the supplementary material (S6). Assessment instruments and topics guides are included as additional files. Transcripts generated and analyzed for the study are available on reasonable request. Anonymized transcripts can be accessed via email request to the corresponding author charlotte.hemingway@lstmed.ac.uk or to library@lstmed.ac.uk quoting the DOI: 10.57978/9eqa-8s37 Transcripts have been stored in LSTM's archive ensuring persistent and long data availability. Transcripts will be shared at the authors discretion with identifiable information redacted to protect the participants' anonymity.

**Funding:** Development and evaluation of the eSPT was funded by the Bill and Melinda Gates Foundation [Grant number INV-024346] through the University of California, San Francisco Malaria Elimination Initiative. The funders had no role in study design, data collection and analysis, decision to publish, or preparation of the manuscript.

**Competing interests:** I have read the journal's policy and the authors of this manuscript have the following competing interests: CH, SG, MC, NL are members of the eSPT development team, as the tool is free to use and distribute, they do not stand to gain financially from the publication of this manuscript.

## Results

Quantitative measures showed that the facilitated eSPT workshop increased participants' knowledge and self-efficacy in question-based entomological surveillance planning. Target users responded positively to the eSPT, reporting high usability scores and satisfaction with the interface. Respondents from academic institutes, central government and international NGOs reported the eSPT to be a useful training tool and believed it could provide substantial efficiencies in the planning process. Further user testing, customizability and compatibility with mobile devices was recommended to enhance the eSPT's usefulness as a planning tool, especially at the local government level.

## Conclusions

Interactive digital toolkits could be an engaging, efficient, and accessible way to build research and surveillance capacity within relevant organizations and local authorities. This is achieved by combining tailored information and guidance, with functions that enable the development of a planning document, in an easy-to-follow stepwise process. To maximize the usability and usefulness of these toolkits, target users must be centered in the design.

## Introduction

Effective entomological surveillance planning is crucial for vector-borne disease control programs. Vector control is the principal method to prevent malaria transmission [1] and has led to significant reductions in malaria incidence across endemic regions [2–5]. However, gains in malaria prevention have stagnated, underscoring the need for tailored vector control that targets local drivers of transmission, and associated entomological surveillance systems that are designed according to the entomological and epidemiological context. Entomological surveillance is essential for understanding vector species, insecticide resistance profiles, and behavioral traits that affect disease transmission and intervention effectiveness. Understanding why and where transmission is persisting is critical to maximizing the effectiveness of vector control and accelerating progress towards malaria elimination.

An Entomological Surveillance Planning Tool (ESPT), developed in 2018 by the Malaria Elimination Initiative (MEI) at the University of California, San Francisco (UCSF) and the University of Notre Dame (UND). The ESPT distills normative guidance from the World Health Organization (WHO) and U.S. President's Malaria Initiative (PMI) into an operational decision-support tool to guide evidence-based vector control.

Originally piloted in Mozambique, Namibia, and Panama, the ESPT has since been embedded within those programs and the resulting data has informed their approaches to vector control. For example, in Panama, the Ministerio de Salud began supporting distribution of bed nets after an ESPT-based surveillance approach demonstrated significant levels of human-vector contact that could be prevented by using bed nets [6]. The ESPT has since been scaled up in sub-Saharan Africa and Southeast Asia and referenced by several national strategic plans, including Kenya's "Malaria Vector Surveillance Operational Guidelines" [7], Uganda's "Entomological Surveillance Framework" [8], and the Democratic Republic of Congo's (DRC) "National Guidelines for Entomological Surveillance and Monitoring and Evaluation of Malaria Vector Control Interventions in the DRC" [9]. The ESPT approach to entomological surveillance planning is also being adopted by the PMI Evolve project with intentions to influence malaria operational planning across PMI-supported countries in sub-Saharan Africa.

To promote adoption of the ESPT's question-based approach to entomological surveillance, an interactive digital toolkit (eSPT) was developed by a team of researchers and software developers (see S1). The eSPT was developed through an iterative user-centered design process: using a range of research techniques, feedback was obtained from users at different stages of product development to ensure their needs and preferences were considered in the design [10, 11]. Two rounds of external usability testing with 10 target users were conducted utilizing the System Usability Scale (SUS), a validated usability questionnaire for software under development. Across the two rounds of external usability testing, the eSPT achieved an average score of 75 out of 100, indicating good usability [12]. Additional open field questions and group discussions were used during an external testing session in Malawi to detect software issues and elicit suggested changes. Feedback obtained from the external testing sessions informed the development of the beta and version 1 of the eSPT. This paper presents a three-country evaluation of version 1 of the eSPT.

This evaluation seeks to understand the eSPT's impact on knowledge acquisition and explore technology acceptability among target users, informing future adoption of interactive digital planning tools in vector control.

## Methods

### eSPT facilitated workshop

The eSPT was evaluated as part of a 2-day facilitated workshop. The workshop was delivered using a combination of short lectures accompanied by practical exercises using the eSPT. The workshop was consistent across the three countries Ethiopia, Malawi, and Mozambique in terms of facilitators, format, and approach; entomological surveillance case studies used in the training material aligned with the participating organizations' research agenda and priority program questions. Participants worked in groups of 2-3, based on their organization and role in entomological surveillance planning, to develop an entomological surveillance plan using the eSPT. An example plan is presented in S2.

### Study type

A mixed-methods, uncontrolled, before and after study design, was deployed to assess the contribution of the eSPT on target users' knowledge, attitudes, and work practices related to entomological surveillance planning. A within-group before-after design was employed with evaluation instruments completed at the beginning and end of the eSPT training course to measure knowledge acquisition (objective 1) and self-efficacy (objective 2). Quantitative data were supplemented with qualitative data to explore the acceptability, perceived utility, and demand for the eSPT among target users (objective 3). Qualitative methods were informed by constructs of technology acceptance extracted from the literature [13], recognizing that assessments of technology acceptability must also systematically consider social, personal and organizational conditions and the subsequent influence of these conditions on technology adoption.

This study design is well-suited for this pilot study, where the primary goal is to assess acceptability and potential impact of the eSPT on knowledge acquisition rather than establish causation definitively. Supplementing quantitative measures with qualitative insights offers a richer understanding of mechanisms driving change and can inform future development and roll out of the eSPT.

### Study sites and population

The study was conducted in three countries: Ethiopia, Malawi and Mozambique, purposely selected due to anticipated variations in exposure to the ESPT, malaria programme priorities and entomological surveillance processes. Between 13 to 18 target users were recruited from each country. Target users were defined as entomologists, researchers, or programme decision-makers

from governmental and non-governmental implementors in vector-borne disease control. Criteria for inclusion in the pilot study as a target user were as follows: involved in planning entomological surveillance activities for vector-borne disease control; and/or involved in management of entomological data for vector-borne disease control; and has access to a PC or Mac computer.

Ethiopia – Ethiopia is implementing its National Malaria Elimination Strategic Plan 2021-2025 which proposes to eliminate malaria in districts with an annual incidence less than 10 by 2025 and the total elimination of malaria from Ethiopia by 2030. Between 2020 and 2021 for the population at risk, estimated incidence decreased from 53 to 46 per 1000, and estimated mortality decreased from 0.12 to 0.10 per 1000 [14]. Recent political and environmental disruptions, compounded by the emergence of insecticide resistance, diagnostic-resistant parasites, and an invasive vector species *An. stephensi*, have driven a 120% increase in malaria cases from 2022 to 2023 [15]. Increases in incidence have not translated to increased mortality, which is attributed to improvements in malaria treatment and use of rapid diagnostic tests for malaria (mRDTs). Insecticide resistance poses a significant threat to vector control interventions deployed in the country, including mass distribution of ITNs and targeted IRS. The 2021-2025 strategic plan includes monitoring and evaluation of insecticide susceptibility, entomological profiles, and intervention effectiveness to facilitate decision making by the National Malaria Elimination Programme (Ethiopia-NMEP).

Entomological surveillance is conducted through collaboration between Ethiopia-NMEP, international funding bodies and local academic research institutes. Armauer Hansen Research Institute and the Ethiopian Public Health Institute act as the research arms of Ethiopia's Ministry of Health. Entomological surveillance activities under the 5-year strategic plan are subcontracted out to the research arms with budget allocated by the Ministry of Health. On infrequent occasions, entomological studies are submitted to the Ministry of Health by the research institutes, proposals are reviewed by Ethiopia-NMEP and an external scientific committee. In total, 26 sentinel sites are run by the Ethiopia Public Health Institute but entomological surveillance is not continuous due to dependency on external funding and no central entomological data repository from sentinel sites and other studies.

Malawi – Malawi's National Malaria Control Program (Malawi-NMCP) is in the process of developing a strategic plan for 2023-2030 with a goal to eliminate malaria by 2030. Between 2016 to 2022, estimated malaria incidence declined from 407 to 208 per 1000 population, and estimated mortality decreased from 23 to 8 per 100,000 population. Declines in incidence and mortality are attributed to the deployment and scale up of multiple malaria control interventions including use of insecticide treated bed nets (ITNs), artemisinin combination therapies (ACTs), Intermittent Preventive Treatment in pregnancy (IPTP), mRDTs, indoor residual spraying (IRS) and more recently the RTS,S malaria vaccine. Malawi is now facing heterogeneous transmission with geographic pockets of high and low transmission, resulting in the Malawi-NMCP using entomological and epidemiological data for targeted intervention decision-making [16].

Entomological surveillance plans are developed through consultation between Malawi-NMCP, technical working groups, international funding bodies, and local academic research institutes. Since 2007, the Malaria Alert Centre (MAC) has implemented malaria vector surveillance on behalf of the Malawi-NMCP, providing research infrastructure (e.g., insectary) and expertise for the planning and implementation of both routine entomological surveillance and targeted surveys. Routine entomological surveillance plans are developed by MAC in consultation with Malawi-NMCP and PMI VectorLink (now PMI Evolve) and cover 18 sentinel sites in 8 districts (with annual budgets), with plans reviewed every 5 years. Research questions for targeted surveys are formulated by senior researchers in consideration of the needs and priorities of the Malawi-NMCP.

Mozambique – Mozambique's National Malaria Control Program (Mozambique-NMCP) is in the process of developing a strategic plan for 2023-2030 in collaboration with the President's

Malaria Initiative (PMI), Clinton Health Access Initiative (CHAI) and other international funding bodies. Between 2015 to 2021, malaria incidence increased from 226 to 328 cases per 1,000 population, mortality decreased from 2,337 to 411 per 10,000 in 2021. Mozambique deploys integrated vector management, i.e., the targeted use of different malaria vector control methods (ITNs, IRS, environmental management), either separately or in combination, to achieve sustainable, ecologically sound, and cost-effective vector control. Entomological surveillance is conducted and the data reviewed on an annual basis to determine the choice of vector control measure(s) [17].

Mozambique-NMCP led entomological surveillance activities are supported by the PMI and the Bill & Melinda Gates Foundation (BMGF) through provision of technical assistance, financial resources and capacity strengthening detailed in annual operational plans [18]. Laboratory analysis of mosquito samples is conducted by Instituto Nacional de Saúde with technical assistance from PMI.

The eSPT workshops were conducted in January 2023 in Ethiopia and Malawi, and May 2023 in Mozambique. Table 1 provides details of vector-borne disease control stakeholders represented at the workshops.

**Table 1. eSPT Workshop Participants.**

| Country | Organization | Role |
|---|---|---|
| Malawi | Malaria Alert Centre, Kamuzu University of Health Sciences | 6x Junior Research Assistants |
| | | 2x Post-Doctoral Fellows |
| | | 2x Senior Research Fellows |
| | PMI Evolve | Senior Research Fellow |
| | Malawi University of Business and Applied Science (MUBAS) | Senior Research Fellow |
| | University of Malawi | Senior Research Fellow |
| | **Total: 13** | |
| Ethiopia | Armauer Hansen Research Institute | 2x Statistician |
| | | Project Manager |
| | | 6x Entomologist |
| | Ethiopian Public Health Institute | 3x Entomologist |
| | Arbaminch University | Biotechnologist |
| | | Entomologist |
| | Ministry of Health | Public Health Officer |
| | | Technical Advisor |
| | Adama Science and Technology University | Entomologist |
| | PMI Evolve | Entomologist |
| | **Total: 18** | |
| Mozambique | Ministry of Health | Technical Advisor |
| | National Malaria Control Program | 3x Entomologist |
| | Provincial Healthcare Service | 2x Entomologist |
| | District Healthcare Service | 2x Entomologist |
| | Manhiça Health Research Center | 2x Entomologist |
| | National Health Institute | 2x Entomologist |
| | Tchau Tchau Malaria | Entomologist |
| | PMI Evolve | Entomologist |
| | World Health Organization | Technical Advisor |
| | PAMCA | 1x Social Scientist 2x Entomologist |
| | **Total: 18** | |

## Data collection

### Objectives 1 & 2

Quantitative data on knowledge and self-efficacy were collected through assessments administered before and after the facilitated workshops using country-specific semi-structured questionnaires (S3a, S3b and S3c). Retention of any knowledge acquired during the facilitated workshops was quantified through administration of the same questionnaires (this time with the self-efficacy component removed) to the same participants at least 30 days following the workshop. The 30-day interval provided a long enough period for new information to be forgotten, and for information gained in the training to still be relevant in the participant's professional role. The 30-day interval was also predicted to provide sufficient time and opportunity for the participant to used the eSPT.

### Objective 3

Qualitative data on acceptability and utility of the eSPT were collected from semi-structured focus group discussions (FGDs) held at the end of the eSPT workshops and follow up individual interviews approximately 30 days later. FGDs and interviews were conducted in the participants' chosen language and facilitated by local researchers BK, TC, EM, LK, MM, YM who were external to the eSPT development team to mitigate against potential response bias, language, or cultural barriers. FGD and interview topic guides (see S4 and S5) were informed by the Extended Technology Acceptance Model (TAM3) (13). FGDs and interviews were audio recorded and transcribed using a de-naturalized approach. Transcripts were translated into English for analysis.

## Data analysis

### Objectives 1 & 2

A dependent samples paired t-test was employed to analyse knowledge acquisition data, which exhibited normal distribution. In contrast, self-efficacy data, which deviated from normality, were analysed using the Wilcoxon signed-rank test. Scores of participants that took only one test were excluded from the analyses. All analyses were performed using statistical package R 4.3.

### Objective 3

Data analysis was informed by a general inductive approach, aligning emerging themes identified in the data with the study objectives and constructs of technology acceptance. Analysis was led by CH in partnership with the local researchers BK, TC, EM, LK, MM, YM. Transcripts from each country were independently coded by the respective in-country qualitative research team and CH, resulting in a data framework and draft narrative. The data framework and draft narrative were shared and discussed with the research team for critical review and collectively revised until agreement was reached on the key findings. The collaborative and iterative approach to the analysis mitigated the risk of research bias and loss of meaning through translation.

### Ethics Statement

Ethical approval for the study was obtained from LSTM Research Ethics Committee (22-019), University of California, San Francisco Institutional Review Board (347594), AHRI/ALERT Ethics Review Committee (PO-50-22), University of Malawi, College of Medicine Research and Ethics Committee (P.08/22/3710), and Comité Nacional de Bioética para a Saúde de

Moçambique (143/CNBS/2022). Written informed consent was obtained from all participants prior to data collection. Study information was either presented in English or the participants chosen language. Minors were not included in the study.

## Results

### Objectives 1 & 2: Knowledge acquisition & self-efficacy

The facilitated eSPT workshop significantly increased entomological surveillance knowledge in participants across the three pilot countries (p = 0.04). Significant knowledge gains were particularly observed in Ethiopia (p = 0.05) and Mozambique (p = 0.03). No statistically significant difference in knowledge acquisition was observed in Malawi (p = 0.41), despite an overall significant increase observed in the results.

Comparison of results from the post-test immediately after the workshop with the 1-month post-test showed that the knowledge acquired during the training was retained (Fig 1). There were no statistical differences between the two tests overall across the three countries (p = 0.72)

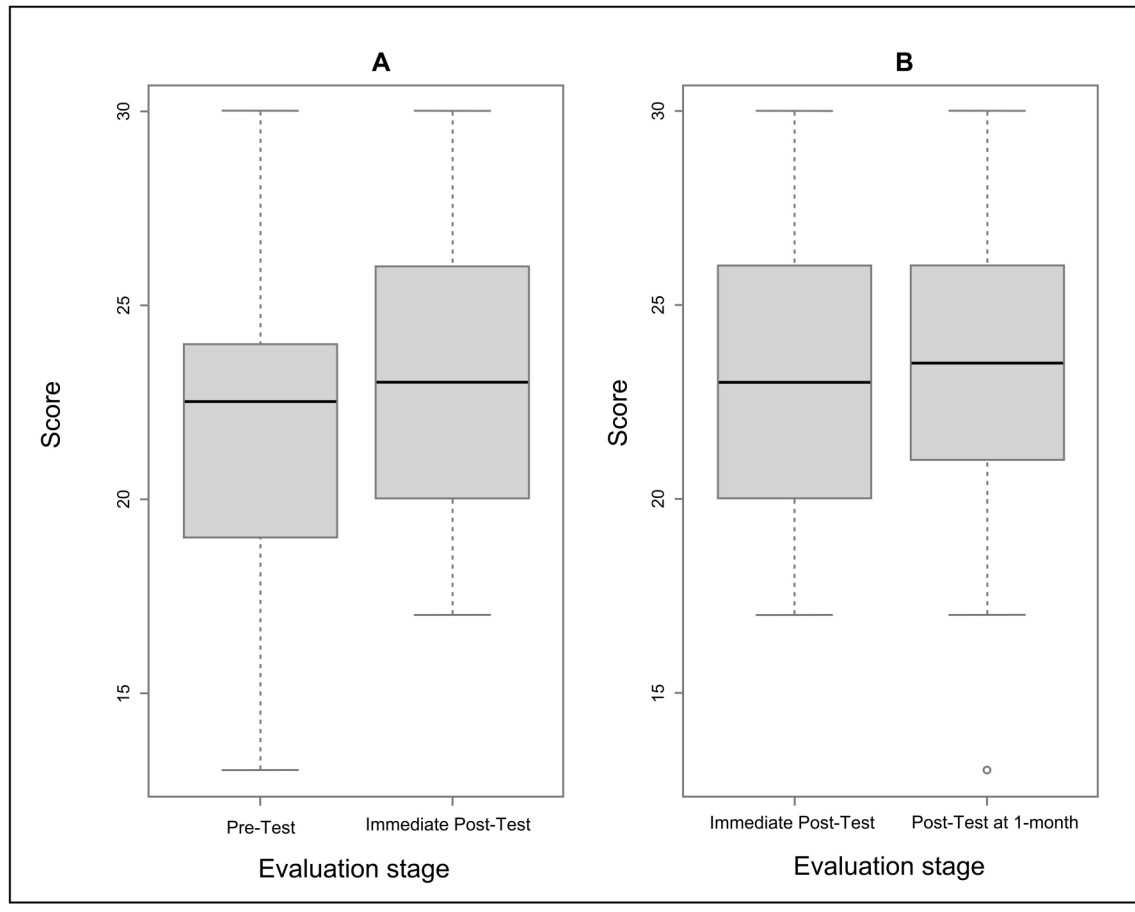

**Fig 1. Box plots showing the contribution of facilitated eSPT workshop on participants' acquisition and retention of entomological surveillance knowledge.** The horizontal line in the box is the median, the boxes represent the interquartile range, the whiskers represent 1.5 times the interquartile range, and dots represent outliers. **A** Participants' entomological surveillance knowledge before and immediately post-training workshop (p = 0.04). **B** Participants' retention of entomological surveillance knowledge at least a month after training workshop (p = 0.72).

or at country level (p = 0.83, p = 0.68 and p = 0.52 for Ethiopia, Malawi, and Mozambique, respectively) (Fig 1).

Participants' self-efficacy to develop entomological surveillance plans increased significantly following the facilitated workshop across the three countries (p < 0.01) (Fig 2).

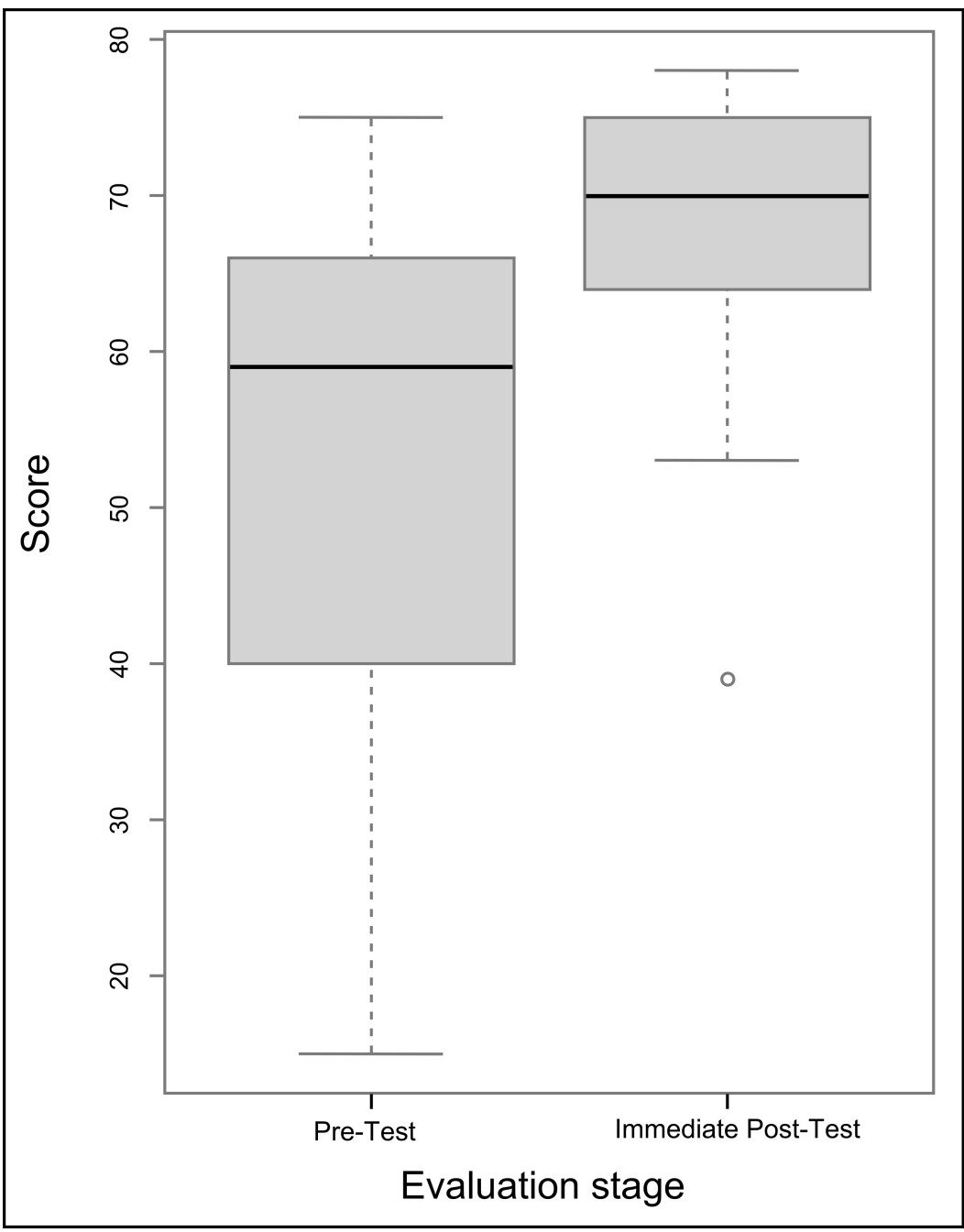

**Fig 2. Box plots showing the contribution of facilitated eSPT workshop on participants' self-efficacy to develop tailored entomological surveillance plans.** The horizontal line in the box plot is the median, the boxes represent the interquartile range, the whiskers represent 1.5 times the interquartile range, and dots represent outliers. A significant difference was found in participants' self-efficacy before and immediately after the training workshop (p < 0.01).

## Objective 3: Technology acceptability

In Ethiopia, two FGDs were conducted (n = 7 and n = 5) with mixed groups from AHRI, EPHI, Arbaminch University, and Ethiopia-NMCP. Five follow-up interviews were conducted, two representatives from Ethiopia-NMCP, two senior researchers from EPHI, and a junior researcher from AHRI. All participants were professionally educated with MSc level qualifications or higher and self-reported to have good computer literacy. Two participants noted that they were very open to new software and technology.

In Malawi, two FGDs were conducted consisting of early career researchers (ECRs) (n = 6) and senior researchers (n = 4) from Malaria Alert Centre, PMI Evolve and MUBAS. Five follow up interviews were conducted, one a retired senior research fellow with extensive experience in entomological surveillance, and four ECRs who support implementation of routine entomological surveillance and targeted surveys for Malawi-NMCP through their roles and studies. All participants were professionally educated with bachelor's degree level qualifications or higher and self-reported to have good computer literacy and felt confident using new software.

In Mozambique, two FGDs were conducted (n = 8 and n = 6) with mixed groups from the Ministry of Health, Serviços Provinciais de Saúde (Provincial Health Services), Direcções Provinciais de Saúde (Directorates of Health DPS), Manhiça Health Research Center, Health National Institute and Tchau Tchau Malaria. Five follow up interviews were conducted, three with Ministry of Health representatives, one from PMI Evolve and one from DPS. All participants were professionally educated with bachelor's degrees or higher and self-reported to have good computer literacy.

## Perception of eSPT workshop

Response to the eSPT workshop was positive across the board. Participants described the workshop as 'helpful' and 'fruitful' in its provision of step-by-step guidance through the entomological surveillance planning process and focus on problem identification and prioritization. The eSPT reportedly provided a user-friendly space in which participants could apply and practice knowledge gained through the lectures and discussions.

Reported limitations of the workshop centered around timing. Some participants felt that 2 days were not sufficient to master the software, or consider the finer details of entomological surveillance planning, and wished to see more time allocated to the group work using the eSPT. Participants recommended aligning the eSPT workshop with in-country planning activities as illustrated by this quote:

*"As there was no specific research problem we identified to use the tools [eSPT] after we had taken this training, I don't think this benefited me well. Because after that training [eSPT facilitated workshop], there was no occasion we had to plan surveillance or research." – Ethiopia_follow up interview*

Participants described how the eSPT workshop had enhanced their knowledge and understanding of minimum essential indicators and mosquito collection methods, enabling them to maximize use of limited resources to collect programmatically relevant data, as illustrated by this quote:

*"I feel like…I now understand the importance of having…a systematic way of doing things. In our case, we collect a lot of data, some of which is not useful to answer our questions. But, in this case, I now know that we need to be direct about what you want based on the budget,*

*and it has also taught me that there are different methods and helped me realize that a single method can help answer a lot of questions, rather than what we usually do, we employ a lot of sampling methods, which I would say are also a redundancy."* – Malawi_follow up interview

### eSPT usability

The eSPT was universally described as easy to navigate and learn how to use; participants were confident they could teach others how to use it. A reported prerequisite to using the eSPT confidently and effectively was a working knowledge of vector borne disease control, entomological surveillance techniques and the associated terminology; although, it was acknowledged that the eSPT provides definitions for much of the technical language used. Some participants believed that with additional training, even those naïve to entomology and vector control could effectively use the eSPT:

*"I think for every step, there is a background knowledge that, if you already have it, you go through that step very fast. Still, if you don't, there are a few… like. For example, with the sampling tools, if you have never heard of any of the sampling tools, reading the descriptions of those 10-15 sampling tools would take you 30 minutes, and then those are very brief descriptions. And if you have never heard of those tools, you already have to seek advice from somebody else with experience in those tools to make informed decisions."* – Malawi_focus group discussion

Participants noted the eSPT's potential to facilitate devolution of planning activities to the provincial or district level, supporting local government worker efforts to adapt research activities to fit the reality of the local environment:

*"Well, I think that it [eSPT] is an opportunity, yes, because in fact any sector has a guiding plan and no matter how much it may be at the central level, when the province arrives we still have to adjust to the reality of the province and using the tool [eSPT] it can also help us to guide our plan at the provincial level. I think it's an opportunity that we can use a lot."* – Mozambique_focus group discussion

However, access to desktop or laptop computers within local governments was reported as an issue that could hinder roll out of the eSPT. Device access was not perceived as an issue at the central government level and in the research institutes. Participants also recognized and showed appreciation for the low computational power required.

Participants from all three countries made multiple requests to make the eSPT compatible with mobile devices. Mobile devices were perceived to be more broadly available, especially when conducting research planning activities 'out in the field,' a phrase used to describe entomological surveillance work conducted away from the research institutes or central government buildings. Participants advised that the eSPT be converted into a web app to provide broader device compatibility and address usability issues relating to software installation restrictions on work/government computers.

There were mixed views on whether participants would be supported by their organizations and/or senior colleagues to use the eSPT. Across all three countries, funding bodies and research institutes, who set standards and documentation formats, were perceived to hold a certain degree of control over software and guidelines used for entomological surveillance planning. Participants noted that the eSPT does not provide a complete protocol in a format suitable for ethical approval or a funding proposal. Rather, it provides an initial scope of

work that would need to be adapted and further developed to meet organizational standards and formats. For these reasons, there was uncertainty among the participants on whether the eSPT would be endorsed by their organization or other important external partners. Participants were optimistic that organizational leaders and/or their immediate superiors would support use of the eSPT given its perceived usefulness in the initial planning stages, but that uptake would hinge on effective and ongoing promotion among key stakeholders. Participants appeared motivated to promote the eSPT within their professional community:

> "The tool is very useful for us, we are saying this not only for lip service but we are sure it is useful. By the way the ministry is not poor; it can communicate with partners for necessary supports. As you all saw Vector Link believed on it and we can provide training. They [EPHI] are the research wings of MoH and if they endorse it, it will be great milestone. I am working as technical advisory there [VectorLink] so I will play my personal role." – Ethiopia_focus group discussion

Participants demonstrated positive attitudes, eagerness, readiness, preparedness, and acceptance towards computers and new software. Participants showed appreciation for the eSPT's aesthetics and found it enjoyable to use.

## eSPT Usefulness

When discussing the usefulness of the eSPT, participants spoke about two use cases, the eSPT as a training tool and a planning tool. As a training tool, members of research and teaching institutes felt the eSPT could be useful for MSc and/or PhD students in developing their proposals and understanding different research methods:

> "PHD students, masters' students, who are trying to develop their own research and planning kind of entomological surveillance, they would love a tool like this; it would make their work much more accessible and shorten the time that they have to sit down and to scratch their heads to think of what indicators to use and what methodologies to apply, you know, possible things.
>
> Whilst those on the higher side, with positions of oversight, would be excited to use it, they would see it as a way that is going to facilitate the actual planning instead of having to do a lot of back and forth providing feedback, questions, comments on the planning of the people that they are trying to supervise or advise. It would make their work easier, in that sense they would be able to look at something tangible..." – Malawi_focus group discussion.

In this case, participants believed the eSPT could help students think in a systematic way and generate efficiencies in the learning and teaching process by providing a framework to build onto instead of a blank page. For senior researchers involved in capacity and career development, the eSPT had job relevance as a teaching tool, helping them to support their students/mentees/employees to transition from delivering other people's research plans to developing plans of their own.

For the more experienced participants, the eSPT reaffirmed existing knowledge, while systemizing the decision-making process, supporting identification of methods to maximize entomological surveillance resources:

> "It systematized things to me. I had been doing it traditionally. I actually know which vector sampling should be applied to which from my university training practices. But it has showed me setting minimum indicator, developing question, achieving the indicators and developing ways to

*achieve the indicators…It systemizes my tasks well and makes life simpler. Even though it is not new knowledge it systemizes the scattered ways here and there. "* – Ethiopia_follow up interview

Participants felt the eSPT enhanced the training experience, providing an interactive and engaging space to learn. Appreciation was shown for the interactive decision trees, facilitating exploration of different methods and indicators:

*"For me, what I saw as useful and advantageous in the tool [eSPT] is also the possibility that we can learn from the tool. For example, when we look at those decision trees, it's like a puzzle. We can try to explore the different possibilities of how to use the entomological indicators to design a certain proposal." – Mozambique_focus group discussion*

Discussion was held on whether use of the eSPT equates to 'cheating' as it collates scientific methodology and expert advice, a job typically assigned to early career researchers. This was often refuted by other participants in recognition of the fact that the eSPT only guides and informs the user and does not make decisions for them, equating the eSPT to web-based search engines. Discussions concluded with the notion that the eSPT does not replace the researcher, but it can make their work easier and their time more productive. It was advised that additional references be included to encourage further reading and understanding of the entomological surveillance techniques.

As a planning tool, participants were very enthusiastic about the eSPT's potential to significantly reduce the time it takes to plan entomological research, some reportedly believing it could reduce the time by several days. Participants described how the eSPT enhanced productivity and 'streamlined' the planning process while helping them maximize limited resources through use of minimum essential indicators and guidance on sampling methods. This in turn would also provide content for resource and protocol amendment justifications, supporting communication with funding bodies and other important partners on why further resources or change is needed or beneficial. Continuing with the theme of communication and collaboration, some participants believed the eSPT had potential to reduce professional disagreements and the consequent delays:

*"…there could be a thousand ways to solve the [vector management] problem. For this, we need research, to do this research we have to plan, to plan the research appropriately, we have not to beat around the bush but to use simple tools. This [eSPT] is exceptionally important. It minimizes a lot of up and downs and idea conflicts and even between scientists. It makes them to have common understanding and to enter to the research directly. So, this saves time and resources..." – Ethiopia_follow up interview*

While the usefulness of the eSPT's word document output was reportedly limited due to organizational format requirements, participants were still able to identify use cases in which it could enable efficient and clear communication of surveillance and research plans and be a helpful tool for funding and ethics applications.

Participants recognized the primary target audience to be national malaria control programs. Depending on the participant's role and experience, the eSPT was either a useful tool to capacitate others in entomological surveillance planning, or a tool that made their job and studies easier through productivity gains. Members of research institutes could see potential in the eSPT beyond its primary target audience:

*"I think, trying to go into the mind of the people that designed the tool, I think it was designed from a more of a malaria programmatic point of view. You know… how to run, like… design*

*for the people at the NMCP kind-of… which is great. We want to see that our malaria vector control programs are successful, and I think it's a tool that is exactly aimed at helping them to achieve that. So, that's very good um… however, I think the software has so much potential that you can use it for your experimental research in the lab. I mean, there is that potential when you look at the tool. There is that potential that it can be used for many other different things." – Malawi_focus group discussion*

To reach this potential, numerous participants advised the content of the tool be expanded to include other vector borne diseases and entomological research activities. Associated with this were perceptions that the name eSPT (Entomological Surveillance Planning Tool) did not accurately reflect the scope of the tool.

*"Currently, at [our research institute], we are looking at sleeping sickness, Schistosomiasis and their corresponding vectors/ hosts including Tsetse and snails. So, through reading the literature and scripts, I have also learnt that [our research institute] was also looking at elephantiasis, so it would be important if eSPT covers these areas so that entomologists should not be limited to malaria alone. They should also have other vector-borne diseases. "- Malawi_follow up interview*

Participants were enthusiastic about the eSPT's potential and envisioned a range of activities and initiatives in which the software could prove useful. Such sentiments were at times caveated with advice to further test the eSPT in a professional work environment to gather evidence on its usefulness outside the facilitated workshop context.

## Intended and actual eSPT use

There were variations in reported intention to use the eSPT within the month following the training across the three countries. In Ethiopia, there was reported intention to use the eSPT 1) at upcoming national surveillance meetings after completion of the new national strategic plan, 2) individually for work related research protocol development, and 3) for student supervision. Participants had reportedly shared the eSPT with colleagues and intended to present the eSPT to other partners involved in the national strategic plan. Reported reasons for not using the eSPT since the workshops were being 'out in the field' and busy with non-planning related tasks.

In Malawi, reported intention to use the eSPT appeared to be driven by a perceived duty to evaluate the eSPT 'on the job' and provide further feedback for software improvement. Further intent to use the eSPT within their professional role was driven by the perceived efficiencies it could provide to the planning process. Participants held expectations for future versions of the eSPT to be released covering other vectors and diseases evidenced by their interpretation of the name 'entomological surveillance planning tool.' Several participants did not view themselves as target users given their extensive knowledge of the research planning process and/or their career stage. Participants were motivated to encourage their colleagues to use the eSPT and were confident they could teach them how to use it. Overall, the eSPT appealed most to students and junior researchers.

In Mozambique, there were reported intentions to use the eSPT to support development of operational plans under the next national strategic plan for malaria control. There were also reported intentions to use the eSPT individually for malaria and non-malaria related research activities with similar calls for the scope to be extended to include other vector borne diseases. Participants had reportedly shared and presented the eSPT to colleagues not at the training.

With regards to reported actual use, most participants did not use the eSPT in the month following the training. While there was a strong intention to use the eSPT, and participants could see several professional benefits, most admitted that they had 'not had chance' or 'got around' to using the eSPT. This speaks to the very specific circumstances in which the software is useful. Reported use of the eSPT in the month following the training was predominantly among junior researchers to 1) think through a new research project, or 2) practice using the eSPT. In Ethiopia, participants reportedly used guidance from the ESPT document to adjust the sample size of an ongoing project given resource limitations.

## Discussion

This study evaluated the influence of facilitated use of the eSPT on user acquisition and retention of entomological surveillance knowledge, confidence to develop entomological surveillance plans and acceptability, and usability of the eSPT in relation to technology adoption. Quantitative measures of knowledge acquisition and retention indicate the eSPT's potential to enhance knowledge of question-based, locally tailored, and cost-effective entomological surveillance planning among relevant professionals. Participants across the three pilot countries were all highly educated professionals, knowledgeable about entomological surveillance and vector control. The modest increase in pre- and post-knowledge assessment scores suggests that the eSPT facilitated workshop contained new information and concepts beneficial to those already experienced in entomological surveillance planning. Even in Malawi, where there was no significant increase in knowledge observed, participants demonstrated appreciation for the systemized way of thinking facilitated by the eSPT and saw its potential as a teaching and mentoring tool, with similar sentiments expressed across the other pilot countries. Early career researchers, MSc and PhD students saw benefit in the productivity gains from the eSPT. The positive response to the eSPT as a learning and teaching tool can be explained in part by the positive affective-motivational states that occurred during the facilitated workshop, in which participants described the workshop as engaging, enjoyable and relevant. Previous research has shown that learning can be enhanced by increasing the prevalence of positive affective-motivational states such as interest, engagement, and enjoyment [19, 20]. In this regard, the eSPT could prove to be a useful tool in efforts to address the lack of skilled in-country entomologists, which has previously been identified as a key gap in the capacity of NMCPs to conduct vector surveillance [21]. It should be noted that the eSPT is not designed to replace traditional forms of education and professional development needed to become a skilled entomologist.

In addition to knowledge change, an individual's sense of self-efficacy has been found to correlate with intended and actual changes in work practices [22]. Facilitated use of the eSPT improved target users' confidence in their abilities related to surveillance planning and data-driven decision making. The eSPT may therefore support individuals and organizations that have not previously planned or implemented entomological surveillance to take on that role. A prominent challenge in malaria control in Sub-Saharan Africa is heterogeneity of transmission, whereby effectiveness of different interventions can vary between different geographical zones. Devolution of health services from national to county/ provincial level provides opportunity to develop and implement locally tailored and contextually-relevant entomological surveillance and vector control interventions. Unfortunately, many local governments across Africa lack the skilled personnel required to manage and implement entomological surveillance activities [23]. Participants of the study saw the eSPT's potential to facilitate devolution of planning activities to the provincial or district level; however, this was caveated with

recommendations to further evaluate the eSPT among local government workers and adapt the tool to be compatible on mobile devices to improve accessibility. On the assumption that baseline knowledge and self-efficacy would be much lower among local government workers compared to the national and research institute representatives recruited for this study, further evaluation of the eSPT among this sub-group of target users would be beneficial. This would also provide opportunity to investigate the long-term impact of self-efficacy on work practices, acknowledging that an individual's confidence in their ability and their actual knowledge and ability are not always equally matched and there may be instances in which use of the eSPT leads to overconfidence [24].

The overwhelming positive response to the eSPT's usability could be a strong predictor for technology adoption. Perceived ease of use has been found to directly influence perceived enjoyment [25], and perceived enjoyment has been identified as an important predictor in intention to adopt certain software products [26]. Development approaches such as user-centered agile software development [10] have been found to improve the quality of the developed product and the usability [27]; suggesting that the approach used to design and develop the eSPT was an important factor in how it was perceived by target users.

Despite participants' views on the usability, perceived enjoyment, and usefulness of the eSPT as a planning tool and reported intent to use it to support entomological surveillance planning activities, in the 30-days following the facilitated workshop there was minimal reported actual use of the eSPT. One explanation for this finding was the niche focus of the eSPT on malaria and mosquitoes. This is likely to negatively impact on adoption of the eSPT as technology that is used infrequently (i.e., once-per-year annual surveillance planning) and only useful in very niche use cases is prone to being forgotten. Further explanation may be found in the reported need for organizational endorsement and support required to use the eSPT for professional tasks. Participants also advised the eSPT be tested in a professional work environment to gather evidence of its usefulness outside the facilitated workshop context. This would suggest that respondents were not convinced of the eSPTs usefulness in their professional role, especially among senior researchers and program officers. Adoption of the eSPT is therefore likely to be dependent on ongoing advocacy and active marketing within relevant networks.

## Limitations

A key limitation of the study was the timing of the facilitated workshops as these did not align with entomological surveillance planning cycles in-country. Timing of the workshops may have negatively impacted on the perceived usefulness of the eSPT. In addition to this, follow-up interviews one month post workshop did not provide a sufficient amount of time to identify any changes in work practices involving use of the eSPT. While target users could see the eSPT's potential to generate significant efficiencies in the planning process, their professional roles and responsibilities often expanded beyond entomological surveillance for malaria control which negatively impacted on the perceived job relevance of the eSPT. Expanding the content to include other vector borne diseases and/or health related surveillance would broaden the usefulness of the software, especially among researchers and academics.

As with all qualitative research, caution should be taken when generalizing the findings to other contexts. However, as there were commonalities in the themes identified across the three pilot countries, the findings may still hold relevance to other similar contexts. The study included highly educated and knowledgeable participants, observed impact of the eSPT on knowledge and reported technology acceptable may be different among target users with lower starting knowledge of entomological surveillance principles and terminology. Given participant's perception of the eSPT as a useful capacity development tool, further evaluation

of the eSPT among participants with less experience in entomological surveillance planning is recommended.

Certain terms used by interview participants did not have a direct English translation which, at times, led to ambiguity in the data during the translation process. To mitigate against the risk of misinterpretation, transcription and translation of interviews not conducted in English was carried out by local researchers fluent in the respective language. The involvement of local researchers familiar with the participants may have influenced their responses. To mitigate against positive response bias, none of the local researchers were involved in the eSPT development and did not have a vested interest in the software.

## Conclusions

This study demonstrates the benefits of the facilitated eSPT workshop on entomological surveillance knowledge acquisition and retention and confidence to develop entomological surveillance plans. Qualitative findings indicate acceptance of the eSPT among key stakeholders and target users and an intention to adopt the software for entomological surveillance planning and training activities.

Findings from this study indicate that interactive digital toolkits could be an engaging, efficient, and accessible way to build research and surveillance capacity within relevant organizations and local authorities. This is achieved by combining information and guidance, with functions that enable the development of a planning document, in an easy-to-follow stepwise process. To maximize the usability and usefulness of these toolkits, target users must be engaged and centered in the design and development process.

## Supporting information

**S1 Text.** S1_eSPT Explainer & Download. Further information about the eSPT software and directions on how to access it.
(DOCX)

**S2 Text.** S2_example eSPT output. An example entomological surveillance planning document generated by the eSPT.
(DOCX)

**S3 Text.** S3a_Malawi Knowledge & Self Efficacy assessment. Survey tool used in Malawi.
(PDF)

**S4 Text.** S3b_Ethiopia Knowledge & Self Efficacy assessment. Survey tool used in Ethiopia.
(PDF)

**S5 Text.** S1 Text. S3c_Mozambique Knowledge & Self Efficacy assessment. Survey tool used in Mozambique.
(PDF)

**S6 Text.** S4_FGD Topic Guide_eSPT ICT Pilot Study. Focus group discussion topic guide.
(DOCX)

**S7 Text.** S5_IDI Topic Guide_eSPT ICT Pilot Study. In-depth interview topic guide.
(DOCX)

**S8 Text.** S6_Knowledge Assessment Responses. Pseudonymized knowledge and self-efficacy response data.
(XLSX)

## Acknowledgements

We would like to acknowledge Bobby Farmer (co-producer and director of EM Studios), Aitor Prado (lead coder), Lindi Harrison (lead artist) for bringing our vision of the eSPT to life. We would also like to acknowledge the participants and their respective organizations, for sharing their time and valuable insights.

## Author contributions

**Conceptualization:** Charlotte Hemingway, Allison Tatarsky, Élodie A. Vajda, Emily Dantzer, Michael Coleman, Neil F. Lobo.

**Data curation:** Steven Gowelo, Mercy Opiyo, Dulcisaria Marrenjo, Mara Maquina, Blessings N. Kaunda-Khangamwa, Lusungu Kayira, Teklu Cherkose, Yohannes Hailemichael, Neusa Torres, Estevao Mucavele, Muanacha Mintade, Baltazar Candrinho, Themba Mzilahowa, Endalamaw Gadisa, Emily Dantzer, Edward Thomsen, Neil F. Lobo.

**Formal analysis:** Charlotte Hemingway, Steven Gowelo, Blessings N. Kaunda-Khangamwa, Lusungu Kayira, Teklu Cherkose, Yohannes Hailemichael, Neusa Torres, Edward Thomsen, Michael Coleman, Neil F. Lobo.

**Funding acquisition:** Allison Tatarsky, Michael Coleman.

**Investigation:** Charlotte Hemingway, Steven Gowelo.

**Methodology:** Charlotte Hemingway, Élodie A. Vajda, Emily Dantzer.

**Project administration:** Charlotte Hemingway, Steven Gowelo, Mercy Opiyo, Dulcisaria Marrenjo, Mara Maquina, Neusa Torres, Estevao Mucavele, Muanacha Mintade, Baltazar Candrinho, Themba Mzilahowa, Endalamaw Gadisa, Allison Tatarsky, Élodie A. Vajda, Emily Dantzer, Edward Thomsen, Michael Coleman, Neil F. Lobo.

**Supervision:** Edward Thomsen, Michael Coleman, Neil F. Lobo.

**Writing – original draft:** Charlotte Hemingway, Steven Gowelo.

**Writing – review & editing:** Mercy Opiyo, Dulcisaria Marrenjo, Mara Maquina, Blessings N. Kaunda-Khangamwa, Lusungu Kayira, Teklu Cherkose, Yohannes Hailemichael, Neusa Torres, Estevao Mucavele, Muanacha Mintade, Baltazar Candrinho, Themba Mzilahowa, Endalamaw Gadisa, Allison Tatarsky, Emily Dantzer, Edward Thomsen, Michael Coleman, Neil F. Lobo.

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
