## [Decision Letter · Decision Letter 0]

12 Jul 2024

Dear Dr. Hemingway,

Thank you for submitting your manuscript to PLOS ONE. After careful consideration, we feel that it has merit but does not fully meet PLOS ONE’s publication criteria as it currently stands. Therefore, we invite you to submit a revised version of the manuscript that addresses the points raised during the review process.

We look forward to receiving your revised manuscript.

Kind regards,

Delfina Fernandes Hlashwayo, Ph.D.

Academic Editor

PLOS ONE

Journal Requirements:

Development and evaluation of the eSPT was funded by the Bill and Melinda Gates Foundation [Grant number INV-024346] through the University of California, San Francisco Malaria Elimination Initiative.

I have read the journal's policy and the authors of this manuscript have the following competing interests: 

CH, SG, MC, NL are members of the eSPT development team, as the tool is free to use and distribute, they do not stand to gain financially from the publication of this manuscript.

We note that one or more of the authors are employed by a commercial company: eSPT development team. 

“The funder provided support in the form of salaries for authors, but did not have any additional role in the study design, data collection and analysis, decision to publish, or preparation of the manuscript. The specific roles of these authors are articulated in the ‘author contributions’ section.”

5. In the online submission form, you indicated that Data underpinning the results of this study are available upon request by email to the corresponding author charlotte.hemingway@lstmed.ac.uk

Transcripts will be shared at the corresponding authors discretion with identifiable information redacted to protect the participants’ anonymity.

Reviewers' comments:

Reviewer's Responses to Questions

**Comments to the Author**

1. Is the manuscript technically sound, and do the data support the conclusions?

Reviewer #1: Yes

Reviewer #2: Yes

Reviewer #3: Partly

2. Has the statistical analysis been performed appropriately and rigorously?

Reviewer #1: Yes

Reviewer #2: Yes

Reviewer #3: I Don't Know

3. Have the authors made all data underlying the findings in their manuscript fully available?

Reviewer #1: Yes

Reviewer #2: Yes

Reviewer #3: Yes

4. Is the manuscript presented in an intelligible fashion and written in standard English?

Reviewer #1: Yes

Reviewer #2: Yes

Reviewer #3: Yes

**Reviewer #1:**  It is a well written manuscript that shows the useful information provided by the eSPT (entomological Surveillance Planning Tool) to improve malaria surveillance. It improved the understanding and self-assurance of professionals engaged in entomological surveillance, even though the participants were already highly educated and experienced. The tool was well-received as a helpful resource for learning and teaching, especially by early career researchers and students, because of its interactive features.

**Reviewer #2: ** Good overall tool

In lines 329 to 360, there are identifiable format changes needed, are plans in place to address?

Can cost savings/benefits be estimated by planning efficiency and time resources utilizing the tool in planning?

What obstacles are there for acceptance/utilization of the tool by country MOH/NMCP and plans to address

**Reviewer #3:**  I consider that it is partially technically sound because the sample they used per country to test the tool is very small.

Regarding statistical analysis I have no comments because it is not my field of experience.

The underlying data if available.

The manuscript is intelligible and the language is adequate.

I think that before thinking about including more vectors for another disease, it would be essential to accompany the design and development of a tool in conjunction with those responsible for using or implementing this tool, in this case they would be those responsible for the malaria program in each country. . This ensures that the process established in each country for the malaria program is reflected in the developed tool and this will make its acceptance and implementation easier.

Furthermore, I believe that the tool should be tested with the personnel who carry out entomological surveillance at different levels, federal, state and local, in order to identify the appropriate perception and information for each level of user and achieve better results. Although this seems difficult when trying to implement it in different countries since each one depends on a defined structure with a financial resource allocated to this activity that varies from one country to another.

However, I consider that as a teaching and practical tool it fulfills its function very well.

Congratulations on your work.

**Do you want your identity to be public for this peer review?** For information about this choice, including consent withdrawal, please see our Privacy Policy

Reviewer #1: No

Reviewer #2: **Yes: ** Robert A Farlow

Reviewer #3: No

---

## [Author Response · Author response to Decision Letter 1]

21 Aug 2024

Dear Dr Delfina Fernandes Hlashwayo,

Thank you for the opportunity to submit a revised version of our manuscript titled ‘Evaluation of a Digital Entomological Surveillance Planning Tool for Malaria Vector Control: Three Country Mixed Methods Pilot Study.’ Please find within our numbered response to points raised by the academic editor and reviewers.

Editor’s comments:

1. Manuscript title page, list of authors, and supporting information files names and citations have been updated in line with journal style requirement.

2. PLOS’ questionnaire on inclusivity in global research has been completed and included in the submission.

3. Statement clarifying the funders had no role in the study added to the financial disclosure.

4. ‘eSPT development team’ is not a commercial company. Authors CH, SG, MC, and NL are all employed by an academic institute as stated under author affiliations. Statement was included under competing interests to ensure transparency over the authors contributions to the design and distribution of the eSPT and to clarify that these authors do not stand to financially gain from uptake and use of the eSPT given its status as a free to use software. Text under competing interest has been updated to avoid confusion.

Competing interests

CH, SG, MC, NL were involved in the design and distribution of the eSPT, as the tool is free to use and distribute, they do not stand to gain financially from the publication of this manuscript.

5. Anonymised responses to the knowledge and self-efficacy assessment have been included as supplementary material (S6). We can not upload focus group discussion and interview transcripts to a public repository or as supplementary material as this would breach compliance with the protocol approved by the research ethics board. The availability of data and materials statement in the manuscript has been edited as follows:

Availability of data and materials

Anonymized response data from the knowledge and self-efficacy questionnaire is provided in the supplementary material (S6). Transcripts generated and/or analyzed during the current study are available from the corresponding author on reasonable request. Assessment instruments and topics guides are included as additional files.

Reviewer 2 comments:

1. In lines 329 to 360, there are identifiable format changes needed, are plans in place to address?

Inclusion of customisable document formats was not feasible within the eSPT development budget and project timeframes. Given the findings of the study, this functionality will be prioritised if additional funding is obtained for further software development. Funding to further develop the eSPT is being pursued by study authors.

2. Can cost savings/benefits be estimated by planning efficiency and time resources utilizing the tool in planning?

Economic evaluation was not within scope of the study, any inclusion of cost saving/benefits resulting from the eSPT would be speculative.

3. What obstacles are there for acceptance/utilization of the tool by country MOH/NMCP and plans to address.

The primary obstacle for acceptance/utilization of the eSPT by government organisations reported by study participants was poor IT infrastructure at district/ provincial level to support devolution of planning activities. This is articulated in the results section.

Reviewer 3 comments:

1. We acknowledge the small sample size of the study, however, note that the sample size was adequate for the statistical methods used and was representative given the small population of people in each country that met the study inclusion criteria.

2. We’d like to thank reviewer 3 for their insightful comments. The authors agree that further development of the software tool should take a more participatory approach with target end users, including target end users at different decision-making levels, enabling the development of more accessible and useful functionality.

We’d like to thank the reviewers for their time and expertise and hope we have adequately addressed the comments raised.

---

## [Decision Letter · Decision Letter 1]

14 Nov 2024

Dear Dr. Hemingway,

Thank you for submitting your manuscript to PLOS ONE. After careful consideration, we feel that it has merit but does not fully meet PLOS ONE’s publication criteria as it currently stands. Therefore, we invite you to submit a revised version of the manuscript that addresses the points raised during the review process.

We look forward to receiving your revised manuscript.

Kind regards,

Muzafar Riyaz, Ph.D.

Academic Editor

PLOS ONE

Reviewers' comments:

Reviewer's Responses to Questions

**Comments to the Author**

Reviewer #4: All comments have been addressed

Reviewer #5: All comments have been addressed

2. Is the manuscript technically sound, and do the data support the conclusions?

Reviewer #4: Partly

Reviewer #5: Yes

3. Has the statistical analysis been performed appropriately and rigorously?

Reviewer #4: Yes

Reviewer #5: Yes

4. Have the authors made all data underlying the findings in their manuscript fully available?

Reviewer #4: Yes

Reviewer #5: Yes

5. Is the manuscript presented in an intelligible fashion and written in standard English?

Reviewer #4: Yes

Reviewer #5: Yes

Reviewer #4: With respect

This research was re-examined in terms of statistics and content.

Please make the following corrections:

The study employs an uncontrolled design, meaning there is no control group to compare the impact of the eSPT workshops. This makes it difficult to determine if the observed knowledge gains and changes in self-efficacy were due to the intervention or other external factors.

While the authors acknowledge the small sample size, it raises concerns about the generalizability of the findings. Additionally, the study only includes highly educated participants with strong prior knowledge, which may not reflect the challenges faced by less experienced or less-educated users.

The follow-up period of only one month may be insufficient to assess the long-term impact of the eSPT on work practices and knowledge retention. Longer-term studies could provide more insights into whether participants continue using the tool and how it influences their work over time.

The workshops did not align with the participants' entomological surveillance planning cycles, which could have negatively influenced their perceptions of the tool’s relevance. This lack of alignment with real-world activities reduces the study's ecological validity.

The eSPT is designed specifically for malaria vector control, limiting its usefulness to researchers or programs dealing with other vector-borne diseases. Some participants indicated that broader applicability would enhance the tool’s value.

Despite the qualitative nature of much of the data, there are attempts to generalize the findings beyond the study sample. The authors caution against generalizing but then draw conclusions that extend beyond the scope of the study.

The involvement of local researchers in data collection was intended to reduce cultural and language biases, but there remains a potential bias due to the participants' familiarity with these researchers, which may influence their responses.

Despite participants' expressed enthusiasm for the tool, the actual reported use of the eSPT was minimal in the month following the workshop. This raises questions about the practical utility of the tool outside the controlled environment of the training sessions.

Depending on the nature of the study, you can benefit from and cite the following articles:

- Knockdown resistance (kdr) associated organochlorine resistance in mosquito-borne diseases (Culex quinquefasciatus): Systematic study of reviews and meta-analysis

- Knockdown resistance (kdr) Associated organochlorine Resistance inmosquito-borne diseases (Anopheles culicifacies): Systematic reviewsstudy

- Knockdown resistance (kdr) Associated organochlorine Resistance in mosquito-borne diseases (Culex pipiens): Systematic study of reviews and meta-analysis

- Knockdown Resistance (kdr) Associated Organochlorine Resistance in Mosquito-Borne Diseases (Anopheles subpictus): Systematic Review Study

Good luck.

Reviewer #5: Evaluation of a Digital Entomological Surveillance Planning Tool for Malaria Vector Control: Three Country Mixed Methods Pilot Study

Abstract

1. The phrase “contributed to shrinking the malaria map” is vague and colloquial for an academic abstract. Use specific language, such as “has led to significant reductions in malaria incidence across endemic regions.”

2. The sentence “an Entomological Surveillance Planning Tool (ESPT) was developed to distil normative guidance into an operational decision-support tool...” blends past and present tenses ambiguously. To keep focus and clarity, revise as “The ESPT was designed to translate normative guidance into an operational tool that supports...”

3. The phrase “users found the software easy and enjoyable to navigate” is subjective and lacks rigorous scientific measurement. It would be scientifically sound to rephrase this based on the specific positive feedback mechanisms observed, e.g., “Users reported high usability scores and satisfaction with the interface.”

4. “A mixed-methods, uncontrolled, before and after study...” is somewhat confusing without further clarification of each method used and their alignment with study goals. A more detailed description, like “A mixed-methods design was employed, combining pre- and post-intervention surveys with in-depth interviews to assess...”,

5. Objectives are sometimes introduced ambiguously. For instance, “to support roll-out of the ESPT’s question-based entomological surveillance planning an interactive digital toolkit, eSPT, was developed.” This lacks clear flow. Streamline objectives by specifying roles and tools concisely: “To facilitate ESPT implementation, an interactive digital toolkit (eSPT) was created to support question-based surveillance planning.”

Introduction

1. The phrase “shrinking the malaria map” is colloquial and too informal for a scientific introduction.

2. The description of ESPT development is long-winded and could be more concise, especially in a background section.

3. Sentences like “An Entomological Surveillance Planning Tool (ESPT), developed in 2018 by the Malaria Elimination Initiative (MEI) at the University of California, San Francisco (UCSF) and the University of Notre Dame (UND), with guidance from a technical working group, distills normative guidance...” are complex and contain multiple clauses. Break down long sentences to avoid overwhelming readers.

4. Details about the user-centered design process and usability testing are long and distract from the primary study aim. Summarise this process in one sentence.

5. While the introduction provides background, it lacks a strong statement about the study's significance and how it contributes to the current body of knowledge. Conclude the introduction with a clear statement of the study’s relevance, such as: “This evaluation seeks to understand the eSPT’s impact on knowledge acquisition and usability, informing future adoption of digital planning tools in vector control.”

6. Phrases like “contextually-relevant” and “question-based entomological surveillance planning” lack clear definitions. Replace with precise language or provide definitions.

Methods

1. The terms “countries,” “sites,” and “locations” are used inconsistently, making it difficult to follow the study structure.

2. The study design, a “mixed-methods, uncontrolled, before and after study,” is briefly mentioned without explaining its relevance or rationale for this approach.

3. It is unclear why certain data, such as the knowledge retention questionnaires, were collected “at least 30 days following the workshop.” Specify the rationale for the 30-day interval.

4. Participant recruitment is described broadly; however, criteria specifics (e.g., what defines a “target user”) are unclear. Refine criteria to clarify who qualified as a “target user.”

5. The use of statistical tests, such as “dependent samples paired t-test or its non-parametric equivalent, Wilcoxon signed-rank test,” could be clearer. Provide context on when each test was applied, such as, “A dependent samples paired t-test was used to analyse normally distributed data; otherwise, the Wilcoxon signed-rank test was applied.”

Results

1. Results Interpretation: Discuss each finding (e.g., “slightly significant” can be replaced with a more specific interpretation) and consider mentioning potential contributing factors for non-significant findings.

2. Use terms like “significantly increased” or “no significant difference observed” instead of “slightly significant.” Clearly state whether a statistical test was parametric or non-parametric.

3. Use of “Overall” and “Country Level” Comparison: Consistently differentiate between overall and country-specific results by specifying these terms in each relevant sentence. For instance, “No significant difference was observed in knowledge in Malawi (p = 0.41), while overall results showed significant retention.”

4. Introduce each quote by paraphrasing or summarizing the point before citing verbatim, making the qualitative data more digestible.

5. Indicate the number of participants for FGDs and interviews at the beginning of each objective’s section to set the context.

6. Conclude each objective’s section with participant recommendations for software improvements, organizational support, or resource needs.

Discussion

1. Statements like "The positive response to the eSPT as a learning and teaching tool can be explained by the positive affective-motivational states..." need more empirical backing or citations beyond the general assertion. Rephrase or add references to justify the claim.

2. The statement "The modest increase in knowledge suggests that the eSPT contained new information..." is vague. Specify the metrics of the "modest increase" to clarify the level of knowledge change and its potential causes.

3. Phrases like "could be a useful tool" and "the eSPT may therefore support..." use conditional phrasing, which is appropriate, but statements regarding the eSPT’s impact on capacity-building or filling gaps in expertise are too generalized.

4. The statement about limited use of the eSPT post-workshop is significant but lacks further analysis. Consider exploring reasons beyond niche focus, such as environmental or organizational barriers. Discuss potential reasons for low engagement beyond niche focus, like accessibility issues, technical challenges, or competing priorities.

**Do you want your identity to be public for this peer review?** For information about this choice, including consent withdrawal, please see our Privacy Policy

Reviewer #4: **Yes: ** Ebrahim Abbasi

Reviewer #5: No

---

## [Author Response · Author response to Decision Letter 2]

9 Dec 2024

A rebuttal letter has been uploaded that responds to each point raised by the academic editor and reviewer(s).

---

## [Editor Report · Decision Letter 2]

16 Dec 2024

Evaluation of a Digital Entomological Surveillance Planning Tool for Malaria Vector Control: Three Country Mixed Methods Pilot Study

PONE-D-24-15126R2

Dear Dr. Hemingway,

We’re pleased to inform you that your manuscript has been judged scientifically suitable for publication and will be formally accepted for publication once it meets all outstanding technical requirements.

Kind regards,

Muzafar Riyaz, Ph.D.

Academic Editor

PLOS ONE

Additional Editor Comments (optional):

The authors have addressed all concerns raised by the reviewers, and I believe it is now suitable for publication.

---

## [Editor Report · Acceptance letter]

PONE-D-24-15126R2

PLOS ONE

Dear Dr. Hemingway,

I'm pleased to inform you that your manuscript has been deemed suitable for publication in PLOS ONE. Congratulations! Your manuscript is now being handed over to our production team.

Kind regards,

on behalf of

Dr. Muzafar Riyaz

Academic Editor

PLOS ONE